# Forest Bathing Is Better than Walking in Urban Park: Comparison of Cardiac and Vascular Function between Urban and Forest Parks

**DOI:** 10.3390/ijerph19063451

**Published:** 2022-03-15

**Authors:** Tsung-Ming Tsao, Jing-Shiang Hwang, Sung-Tsun Lin, Charlene Wu, Ming-Jer Tsai, Ta-Chen Su

**Affiliations:** 1The Experimental Forest, College of Bio-Resource and Agriculture, National Taiwan University, Nantou 55750, Taiwan; soutp@yahoo.com.tw (T.-M.T.); pine88854@gmail.com (S.-T.L.); 2Institute of Statistical Science, Academia Sinica, Taipei 11529, Taiwan; hwang@sinica.edu.tw; 3Institute of Environmental and Occupational Health Sciences, National Taiwan University College of Public Health, Taipei 10055, Taiwan; 4Global Health Program, National Taiwan University College of Public Health, Taipei 10055, Taiwan; charlenewu@ntu.edu.tw; 5School of Forestry and Resource Conservation, National Taiwan University, Taipei 10617, Taiwan; 6Department of Environmental and Occupational Medicine, National Taiwan University Hospital, Taipei 100229, Taiwan; 7Divisions of Cardiology, Department of Internal Medicine, National Taiwan University Hospital, Taipei 100229, Taiwan

**Keywords:** cardiovascular function, blood pressure, walking exercise, urban park, forest bathing

## Abstract

Forest bathing is beneficial for human health. To investigate whether walking in forest or urban parks affects cardiovascular functions (CVFs), the present study was conducted in five forest trails in the Xitou Experimental Forest and in five urban parks in Taipei city. We recruited 25 adult volunteers for an observational pilot study in forest parks (n = 14) and urban parks (n = 11). CVFs were assessed by measuring the arterial pressure waveform using an oscillometric blood pressure (BP) device. The baseline and paired differences of systolic BP (SBP), central end SBP, heart rate, left ventricle (LV) dP/dt max and cardiac output in participants were lower before and after walking in a forest park than those in an urban park. In addition, the systemic vascular compliance and brachial artery compliance of those who walked in a forest park were significantly higher compared with those in an urban park. Linear mixed models demonstrated lower levels of SBP by 5.22 mmHg, heart rate by 2.46 beats/min, and cardiac output by 0.52 L/min, and LV dP/dt max by 146.91 mmHg/s among those who walked in forest compared to those in an urban park after controlling covariates. This study provides evidence of the potential beneficial effects of walking exercise in forest parks on CVFs.

## 1. Introduction

Cardiovascular diseases (CVD) are the leading causes of morbidity and mortality worldwide [1]. The increasing rate of CVD has become a major focus of epidemiological studies, public health policies, and prevention studies. Researchers have reported that multiple factors increase the risk of developing CVD, such as hypertension, dietary habits, smoking habits, genetics, diabetes, environmental pollution, and psychosocial factors [2,3,4]. In addition to major CVD risk factors, the development of CVD is strongly connected to lifestyle factors such as psychosocial stress, unhealthy dietary habits, use of tobacco, and physical inactivity. More than three-fourths of all CVD mortality may be preventable with adequate changes in lifestyle [5]. Lifestyle changes, such as exercise habits, are one of the important preventive measures against CVD, and have been supported by many epidemiological studies [6,7].

Exercises that focus on endurance, resistance, and slow-movement such as Tai-Chi, aerobics, and walking, are recommended as a nonpharmacological therapy for patients with CVD. Exercise training is a well-established adjuvant therapy in diseases such as chronic obstructive pulmonary disease and congestive heart failure [8,9,10]. Many studies have demonstrated that exercise may provide health benefits by reducing blood pressure (BP) and pulse rate as well as by its effects on cardiovascular and metabolic parameters [6,7,11]. Cardiovascular alterations that occur with aging include decreased maximal oxygen uptake and maximal cardiac output [7]. Effects of exercise on cardiac function are relatively well described in young and older subjects [7,12]. Nevertheless, the researchers in the studies cited above investigated the effects of exercise on human cardiovascular metabolic parameters. However, there is limited information regarding exercise habits on cardiovascular hemodynamics [13], particularly investigating walking exercise in a forest environment.

Forest bathing may provide potential health benefits, such as increasing levels of natural killer cell activity and enhancing human immune function [14,15]. Such benefits may be due to the biogenic volatile organic compounds (BVOCs) emitted by trees [16,17,18]. Additional benefits from forest bathing include improved psychological wellbeing, enhanced parasympathetic nerve activity, and decreased sympathetic nerve activity in the human body [19,20]. Many possible therapeutic uses for forest bathing are utilized against COVID-19 in current global events [21,22]. BVOCs could be useful in the prevention and/or treatment of COVID-19 [23]. Forest bathing, accompanied by walking in forests, is considered a beneficial practice against COVID-19-related disorders. Forest environments certainly provide medicinal potential for human health [24,25,26]. The anti-COVID-19 potential of some houseplant-emitted volatile compounds can be exploited in ‘indoor forest bathing’ approaches for immunity boosting and health protection [27].

This study was designed to investigate whether a forest park setting is better than that of an urban park, as well as the underlying mechanism of cardiovascular functions (CVFs) on walking exercise in five forest trails compared with those in five urban parks. In addition to comparing the cardiovascular hemodynamics before and after walking exercise for each participant intra-individually, we also measured the forest and urban environmental air quality simultaneously during the study period.

The aims of our investigation were to (1) compare the measurements of BP, CVFs in participants who engaged in walking in the forest (forest bathing) compared to those walking in urban parks, and (2) demonstrate that forest parks have better air quality compared to that in urban parks, Taipei City, Taiwan.

## 2. Materials and Methods

### 2.1. Design and Participants

The present study was designed with the aim of determining the health effects of walking exercise on CVFs in a forest environment compared with those in an urban park environment. We recruited 25 adult volunteers for an observational pilot study of forest park and urban park. Participants with a mean age of 54.21 years (n = 14) in forest park and 42.36 years (n = 11) in urban park were recruited for this study. All participants provided informed consent before undergoing cardiovascular health examinations. The health examinations of participants were conducted from 21 to 26 June 2016, in National Taiwan University (NTU) Xitou Experimental Forest, Nantou County, Taiwan, and urban park examinations were conducted from 22 to 23 August 2016, in Taipei City, Taiwan.

Fourteen participants joined a 6-day/5-night forest walking exercise program in Xitou forest from 21 to 26 June 2016. During the forest walking, all participants had to maintain dietary control and walking exercise. All participants had to undergo CVFs examinations in the morning at a wood house before the forest walking. The participants walked for 1.5 h in a *Phyllostachys edulis* forest field as a forest walking exercise. Then, all participants came back to start at the wood house and undergo CVFs examinations again after taking a rest for 20 min. The full walking distance was approximately 2.4 km. All participants walked on five different forest trails (*Japanese cedar* forest, *Taiwan red cedar* forest, greensward, *Taiwania cryptomerioides* forest, and *Phyllostachys edulis* forest) for 1.5 h in the morning in NTU Xitou Experimental Forest. On the sixth day, after the health examinations were completed, all participants finished forest walking exercise and returned to Taipei City.

The health examinations of participants in urban parks in Taipei City were conducted from 22 to 23 August 2016. Of the 11 participants who were enrolled for the urban park examination, four also participated in the forest walking exercise program in Xitou forest. During the urban park walking exercise, all participants had to maintain dietary control and walking exercise. On the first day, all participants underwent CVFs examinations in the morning at the 228 Peace Park before walking in urban park. The participants walked for 1.5 h in Memorial Park as a walking exercise. Then, all participants finished urban park walking and underwent CVFs examinations again after a 20-min break. All participants went to the Chiang Kai Shek Memorial Hall Park and underwent CVFs examinations before and after walking exercise in urban park. In the afternoon, all participants went to the Pao An Temple Park and underwent CVFs examinations before and after walking exercise in urban park. On the second day, all participants had to go to the Sun Yat Sen Memorial Hall Park in the morning and the Long Shan Temple Park in the afternoon for urban park examinations.

This study was approved by the 37th meeting (30 January 2013) of the research ethics committee of the NTU Hospital, Taipei City, Taiwan. All participants provided written informed consent before undergoing a series of detailed examinations and questionnaires.

### 2.2. Site Descriptions of the Forest and Urban Park Environments

The Xitou experimental forest site of the NTU covers 2349 ha, mainly populated with natural hardwood forest and some plantations containing predominantly conifers. The forest ranges in elevation from 500 to 2025 m. The climate is subtropical, with an average annual rainfall of 2590 mm between 1941 and 2010. The average annual air temperature and relative humidity from 2011 to 2015 were 17 °C and 88%, respectively, according to the Xitou monitoring station of the Taiwan Central Weather Bureau (TCWB). The five sites of forest exercise training were *Phyllostachys edulis* forest, *Japanese cedar* forest, *Taiwan red cedar*, greensward, and a *Taiwania cryptomerioides* forest located approximately 1150–1300 m within the Xitou forest (Figure 1a and Figure 2).

Taipei City is the largest metropolitan area in the country. The urbanized area covers about 271.8 km^2^ with estimated population of 2,597,000. Taipei City has a monsoon-influenced, humid, subtropical climate in which summer is hot, humid, and accompanied by occasional heavy rainstorms and typhoons. According to the TCWB, the annual average air temperature and relative humidity from 1991 to 2020 was 23.5 °C and 74.8%, respectively. Figure 1b and Figure 2 display the location of the sites of five famous parks in Taipei city, Taiwan including the 228 Peace Park, Chiang Kai Shek Memorial Hall Park, Sun Yat Sen Memorial Hall Park, Long Shan Temple Park, and Pao An Temple Park.

The 228 Peace Park in Taipei City was originally established in 1900 as Taihoku New Park, during the Japanese colonial period. It occupies 71,520 m^2^ with a green area of 54,215 m^2^ (greening rate 75.8%). Of the 60 kinds of trees in this park, the top three dominant trees and their count are *Bischofia javanica* (214), *Ficus microcarpa* (157), *Chrysalidocarpus lutescens* (97). The Chiang Kai Shek Memorial Hall Park in Taipei City is 250,000 m^2^ in area, with green area 117,340 m^2^ (greening rate 46.9%). There are 17 kinds of trees in this park. The top three dominant trees and their count are *Araucaria cunninghamii* (524), *Cerasus serrulata* (300), *Ficus microcarpa* var. *crassifolia* (184). The Sun Yat Sen Memorial Hall Park in Taipei City is 115,702 m^2^ in area, with a green area 63,115 m^2^ (greening rate 54.5%). There are 32 kinds of trees in this park. The top three dominant trees and their count are *Melaleuca leucadendron* L. (199), *Ficus microcarpa* L.f. (138), *Cerasus serrulata* (93). The Long Shan Temple Park in Taipei City is 13,618 m^2^ in area, with green area 3616 m^2^ (greening rate 26.6%). There are 11 kinds of trees in this park. The top three dominant trees and their count are *Liquidambar formosana* (34), *Koelreuteria elegans* (21), *Cinnamomum camphora* (20). The Pao An Temple Park in Taipei City is 5913 m^2^ in area, with green area 2569 m^2^ (greening rate 43.4%). There are 11 kinds of trees in this park. The top three dominant trees and their count are *Ficus microcarpa* L.f. (32), *Juniperus chinensis* L. var. *kaizuka* (10), and *Araucaria cunninghamii* (3).

### 2.3. Exposure Assessments

The instruments used for environmental monitoring were real-time recording on the five forest trails and in the five urban parks. The concentration and size distribution, as well as real-time mass concentration of particulate matter (PM; PM_10_, PM_2.5_, PM_1_), were monitored using a DustTrak aerosol monitor (model 8533; TSI Inc., Shoreview, MN, USA). The total volatile organic compounds (TVOC) concentration was measured with a ppbRAE3000 photoionization detector (RAE Systems Inc., San Jose, CA, USA). The concentrations of CO and CO_2_, temperature, and relative humidity were monitored using an IAQ monitor (model 2211; Kanomax, Andover, NJ, USA).

### 2.4. Cardiovascular Function Assessments

The internet was used for data transmission from the data collection site to a central analysis center, and BP was determined by pressure waveform changes according to Bernoulli flow effects. Vascular compliance and peripheral resistance of the brachial artery were derived by incorporating the arterial pressure signals from a standard cuff sphygmomanometer using a physical model. This method was previously validated against invasive [28] and non-invasive measurements [29]. Brachial artery compliance (BAC) was calculated at mean arterial pressure by theoretical design [28]. Brachial artery distensibility and resistance were derived from the modified formula of BAC, which was associated with cardiovascular risk factors in healthy young adults in the Bogalusa Heart Study [30,31]. This method has also been used to derive other hemodynamic parameters, including cardiac hemodynamics, such as the maximum rate of left ventricular pressure rise (LV dP/dt max), stroke volume (SV), and cardiac output [32] were also proved its application in our recent study [33].

### 2.5. Statistical Analysis

Means ± standard deviation (SD) of continuous variables between groups were compared using the Student’s two-tailed *t*-test or the Mann–Whitney U-test if not in normal distribution. The analysis of variance was used to analyze the differences among group means. Environmental factors, including PM, TVOC, CO, CO_2_, temperature, and humidity in five forest parks and urban parks were compared. The goal of the analysis was to test for the mean differences and to quantify them. We used the paired *t*-test to evaluate each individual’s mean difference of cardiac and vascular hemodynamics measured before and after walking exercise in five forest parks and five urban parks. Regarding CVFs changes after walking in different environments, the differences between in the forest and urban environments were tested by using two sample *t*-test. All statistical analyses were performed with SPSS statistical software (version 19; IBM SPSS Statistics, 2010, Chicago, IL, USA).

Linear mixed-effects models were used to estimate the changes on CVFs after walking (forest bathing) in the Xitou Experimental Forest compared to those changes of cardiovascular functions after walking in urban parks. The individual characteristics was set as random effects, and estimated by restricted maximum likelihood, and fixed effects were model-based after controlling the categories of age, gender, BMI, and hypertension status.

## 3. Results

General characteristics of study participants in forest and urban parks are summarized in Table 1. The mean age of participants in forest and urban parks were 54.2 and 42.36 years, respectively. The enrollees in forest and urban parks were mostly males—71.43% and 72.73%, respectively. The prevalence of hypertension (140 mmHg in systolic BP or 90 mmHg in diastolic BP) in forest and urban parks was 28.57% and 36.36%, respectively. Diabetes mellitus in adults accounted for 28.57% of the participants, and 42.86% were diagnosed with hyperlipidemia in forest parks. Overall, the demographic characteristics of study participants in forest and urban parks were not significantly different.

### 3.1. Environmental Characteristics

The comparison of ambient air quality monitoring between forest and urban parks are presented in Table 2. Levels of PM_1_, PM_2.5_, PM_10_, total PM, TVOC, CO, and temperature were significantly lower in the forest environment than in urban parks. There were significant differences in air pollution between forest and urban parks. However, the relative humidity in forest parks was significantly higher than in urban parks.

The environmental monitoring results in five different forest trails are presented in Table 3. Based on analysis of variance, there were significant differences among the five different forest trails according to results of the *F*-test of analysis of variance. The PM_1_, PM_2.5_, PM_10_, and total PM levels, TVOC, CO_2_, temperature, and relative humidity were significantly different among the five forest trails. However, there were no significant differences in CO level among the five different forest trails.

Ambient air quality monitoring data among the five urban parks are presented in Table 4. There were significant differences among the five urban parks. The PM_1_, PM_2.5_, PM_10_, and total PM levels were significantly lower in Long Shan Temple Park. However, 228 Peace Park had significantly higher levels than other parks. There was significantly better air quality in TVOC, CO, and CO_2_, lower temperature, and higher relative humidity in the Xitou forest environment compared to those of urban parks.

### 3.2. Cardiovascular Function

Brachial systolic BP (SBP), central end-cSBP (cSBP), and pulse pressure (PP) among forest participants were significantly lower after walking exercise in forest parks (Table 5). Cardiac function including heart rate (HR), LV dP/dt max, LV contractility, cardiac output, cardiac index (CI), SV, and SV index among forest participants were significantly lower after walking exercise in forest parks. Systemic vascular compliance (SVC) among forest participants was significantly higher in forest parks. Brachial artery resistance (BAR) among forest participants was significantly lower after walking exercise in forest parks. However, among urban participants, there were no significant differences after walking exercise in urban parks.

Comparison of CVFs changes after walking exercise in forest and urban parks showed in Table 6. The effects after walking exercise among forest participants in forest parks are as follows: Decreased 3.78, and 4.62 mmHg in the SBP and cSBP, respectively; decreased 1.36 and 5.29 mmHg in the mean artery pressure and PP, respectively; and decreased 1.3 beats/min, 97.18 mmHg/s, and 0.25 L/min in the HR, LV dP/dt max, and cardiac output, respectively. There were statistically significant differences between the crude effects in forest and urban parks.

In order to estimate the health effects on CVFs after walking exercise, linear mixed-effects models were used to adjust random effects of individual characteristics and controlling factors of age, gender, BMI, and hypertension status. The estimated effects of forest park vs. urban parks using linear mixed models were shown in Table 7. Brachial SBP and cSBP of participants in forest parks were significantly lower than that in urban parks. The same we found that the heart rate, LV dP/dt max, LV contractility, cardiac output, CI, SV, and SV index in forest parks were significantly lower than that in urban parks. Linear mixed models showed lower levels of SBP by 5.22 mmHg, heart rate by 2.46 beats/min, cardiac output by 0.52 L/min, and LV dP/dt max by 146.91 (mmHg/s), respectively, while those walking in the forest compared to those in urban parks after controlling covariates.

## 4. Discussion

This study is the first in travel medicine to show the potentially beneficial health effects, in terms of CVFs, after walking exercise of participants in forest parks compared to those measured in urban parks. This study provided a novel finding of concurrent CVFs evaluation and real-time environmental monitoring, which showed that improved air quality is accompanied by a better cardiovascular response to walking exercise in forest parks than in urban parks. Our previous study showed that those living in a forest environment may demonstrate improved cardiovascular health, such as significantly lower levels of total cholesterol, low-density lipoprotein cholesterol, fasting glucose, and carotid intima-media thickness than those living in an urban environment [34].

Studies regarding certain activities in a forest environment, such as breathing, relaxation, meditation, and walking exercise have shown beneficial health effects [35]. In the present study, we found that walking exercise has more significant effects on CVFs with exposure to forest parks. SBP and cSBP of participants in forest parks were significantly lower than those in urban parks. It is well known that BP is influenced by multiple factors, relative contributions are probably quite different from person to person. For example, lack of sleep at night, anxiety, having noise within the measured environment, not taking rest before the measurement, and lifestyle factors (e.g., diet, coffee, smoking, alcohol, and exercise) affect BP components. Our findings are consistent with previous studies that exposure to a forest environment by walking or even just staying in the forest environment can decrease BP, cortisol, HR, and sympathetic nerve activity [35,36].

Many studies have confirmed that participants with the widest PP have the greatest risk of mortality associated with CVD [37]. PP has also been reported to be a significant indicator of myocardial infarction [38]. Furthermore, PP is a strong indicator of cardiovascular risk even among normotensive persons [39]. Our study shows that participants’ PP and HR before, after, and paired difference after walking exercise in forest parks were lower than those in urban parks. This study also indicates that exposure to forest environments might decrease PP and HR levels in humans. On the basis of these studies, walking exercise in forest parks, as compared with that in urban parks, can be considered more effective in providing relaxation. Cardiac function, including dP/dt max, LV contractility, cardiac output, and SV, decreased significantly in participants in forest parks compared with those in urban parks. This implies that in humans, cardiac burden is reduced after walking exercise in a forest park. Many studies have shown that contact with a forest environment might make people calmer and more relaxed than in an urban area [40]. Exposure to a forest environment is now receiving increased attention for its effects of reducing stress and providing a feeling of relaxation. Intra-individual paired difference comparisons of SVC, BAC, and BAR before and after walking exercise of participants in forest parks had statistically significant differences from the corresponding differences in urban parks.

Assessments of the impact on cardiac function before and after exercise training are very important for hypertensive patients or general people to improve health. Many studies have reported that exercise training can provoke important changes in the CVFs in humans and animals with hypertension [41,42,43,44]. Study reported that low-intensity exercise training, which is performed on a motor treadmill for 60 min, 5 times per week for 18 weeks, at 55% maximum oxygen uptake (VO_2max_), can significantly decrease BP, HR, and cardiac output in hypertensive rats than in a sedentary condition or a high-intensity exercise training (88% VO_2max_). The decrease in cardiac sympathetic tone in low-intensity exercise trained in hypertensive rats may cause hemodynamic changes (decreased BP, HR, and cardiac output), consequently attenuate the cardiovascular load in hypertension [42]. Study reported that low-intensity exercise training in older hypertensive patients, who were required walking at home for 60 min 3 times per week for 9 months at 50% VO_2max_, revealed a significant reduction in arterial BP, HR, and cardiac output after training [43]. In the present study, walking approximately 2.4 km for 1.5 h in forest trails has more significant reduction on cardiac and vascular workload in a forest environment compared to that in urban parks. Particularly, HR, LV dP/dt max, LV contractility, cardiac output, and SV before and after exercise of participants in forest parks were significantly lower than those of participants in urban parks.

The participants recruited for the observational pilot study all lived in Taipei City and walk regularly in urban parks, with an average walking time of 1.5 h/day. However, the participants in the urban park walking group did not demonstrate improved CVFs. CVFs of participants who walk regularly in urban parks may be affected by traffic air pollution [45,46], sports type and exercise intensity [47], higher ambient temperature in summer [48], and urban heat island effect [49]. The noise levels in the two different environments are different. The city noises (e.g., road traffic noise and engineering construction) exposures engender physiological reactions of stress [50]. Based on the above discussion, we considered inappropriate urban weather conditions, lack of park’s greenness or inadequate tree canopy coverage in urban area might attenuate the beneficial healthy effects of urban parks on CVFs. It is well known that increasing tree canopy coverage or park’s green area can reduce ambient air pollution and temperature in urban area. Thus, improvements in urban green spaces and canopy coverage are a very important issue. Urban forest or afforestation in the city should be encouraged to increase the urban greenness.

There are some possible explanations in our findings to understand cardiovascular phenomenon in forest parks. First, HR, LV contractility, cardiac output, and SV in forest parks decreased after walking exercise, indicating that physiological relaxation on reduction of cardiovascular activity in human body [51,52,53]. Second, exposure to forest environments might enhance the immune response of NK cells (CD3^−^/CD56^+^) and activating NK cells (CD3^−^/CD56^+^/CD69^+^) in humans. The percentage of NK cells was significantly higher in the forest group (19.5 ± 9.1%) than in the urban group (16.4 ± 8.4%). After the five-day in Xitou forest bathing, forest environments and plant-released BVOCs inhaled by walkers, the percentage of activating NK cells of walkers increased significantly from 0.83 ± 0.39% to 1.72 ± 0.1% in our previous studies [15]. Third, our recent studies showed that long-term traffic-related air pollution, concentrations of particulate matters and NO_2_ in Taipei City were closely and independently associated with inflammation, and thrombotic biomarkers [54] Thus, better air quality in forest environments may improve cardiovascular health [34]. Forth, better green landscapes in forest environment may benefit CVFs [55,56,57].

This study has several strengths. The strength of this study demonstrates the consistent health effects of walking exercise in different forest trails on BP components, cardiac function and hemodynamic parameters simultaneously compared with those in urban parks. This study corroborated that the findings for before and after walking exercise, and the paired difference between these cardiovascular hemodynamics revealed the significantly better health effects in forest parks. The unique findings of this study also can be supported by concurrent measurements of environmental parameters, including lower air pollution parameters and higher levels beneficial phytoncides.

However, our study had some limitations. First, our sample size was relatively small, which may cause true effects on the CVFs changes to be undetected. However, the CVFs measurements of the participants were taken in both hands and in five forest trails or five urban parks. With the increased number of measurements, we had a higher chance to detect differences in CVFs changes between walking in forest park areas and urban park areas using linear mixed-effects models. Second, the age, gender, and diseases of our participants in forest and urban parks not evenly distributed. However, Table 1 showed that these characteristics of the two groups were not significantly heterogenous. We also fitted the observed CVFs changes after walking exercises with linear mixed-effects models to adjust potential confounding factors, especially age, gender, BMI and hypertension. Third, there was no control group walking through the urban area to investigate if the parks or forests improve the health condition in comparison to urban area. Fourth, the air temperature and relative humidity in Xitou Experimental Forest and the urban parks were statistically significantly different (10 °C and 30%, respectively). The different weather conditions, including air temperature, relative humidity, atmospheric pressure and wind speed, that occurred during the research in urban and forest parks have not been considered.

Fifth, although our results indicate positive health effects of walking exercise in forest parks compared with urban parks, we could not infer that the health effects are attributable to BVOCs, because BVOCs from tree leaves in forest environments were not investigated in this study. We believe that levels of BVOCs with healthy potential inhalable during forest bathing are higher in forests than urban parks. The health benefits of BVOCs found in forest environments and the mechanisms mediating the effects of phytoncides on the cardiovascular system are very important in public health and environmental sciences. Perspectives of this study also indicated that further studies designed to demonstrate the potential role of phytoncides on CVFs are warranted.

## 5. Conclusions

This study indicates the potential health effects on CVFs of participants who walked in a forest environment. Our results support that a better urban design with increased urban green area and canopy coverage, as well as afforestation in urban cities, may improve the health of citizens. The government should actively develop parks and recreational areas through urban greening plans. Conducting a large-scale cohort study with more participants and investigating other mechanisms including human bioclimatology, and sympathetic/parasympathetic functions, are warranted in future.

## Figures and Tables

**Figure 1 ijerph-19-03451-f001:**
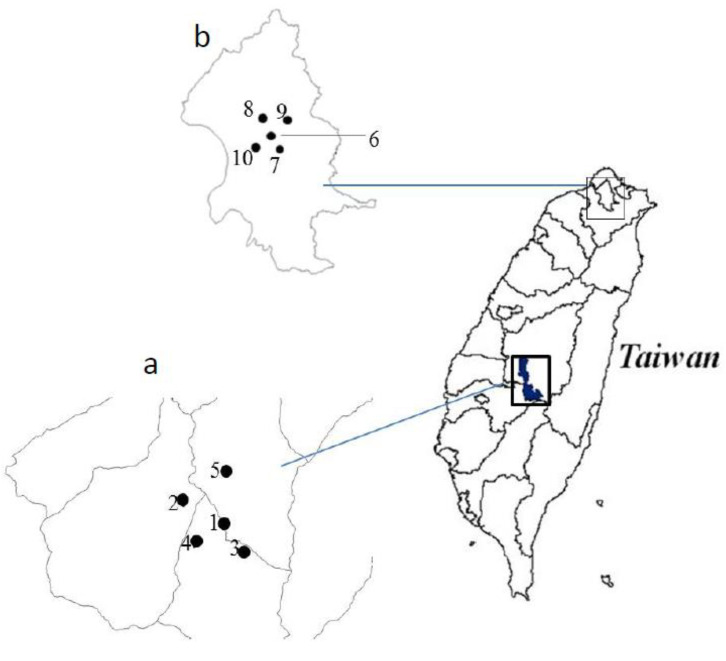
Location of study plot in forest trails and urban parks. The five forest trails in NTU Xitou Experimental Forest, Nantou County, Taiwan (**a**) and urban parks marked (**b**) in Arabic numerals were (1) *Phyllostachys edulis* forest, (2) *Japanese cedar* forest, (3) *Taiwan red cedar* forest, (4) greensward, (5) *Taiwania cryptomerioides* forest; the five urban parks in Taipei City, Taiwan (6) 228 Peace Park, (7) Chiang Kai Shek Memorial Hall Park, (8) Pao An Temple Park, (9) Sun Yat Sen Memorial Hall Park, and (10) Long shan Temple Park.

**Figure 2 ijerph-19-03451-f002:**
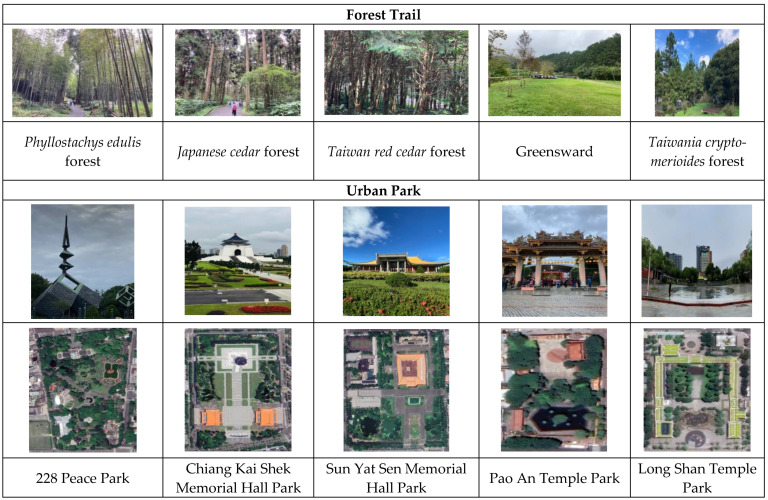
Photos of the five forest trails in NTU Xitou Experimental Forest, Nantou County, and urban parks in Taipei City, Taiwan.

**Table 1 ijerph-19-03451-t001:** The characteristics of 25 participants in forest and urban parks with walking exercise.

Variables	Forest Park		Urban Park		*p*-Value
	N/Mean	%/SD	N/Mean	%/SD	
Gender					
Female	4	28.57	3	27.27	1.0000
Male	10	71.43	8	72.73	
Age (Continuous)	54.21	15.53	42.36	19.50	0.2491
Age (Categorical)					
22–35	3	21.43	6	54.55	0.1153
54–73	11	78.57	5	45.45	
BMI (Continuous)	24.07	3.36	23.34	2.93	0.5287
BMI (Categorical)					
<24	7	50	7	63.64	0.6887
≥24	7	50	4	36.36	
Hypertension					
No	10	71.43	7	63.64	1.0000
Yes	4	28.57	4	36.36	
Diabetes mellitus					
No	10	71.43	11	100	0.1052
Yes	4	28.57	0	0	
Hypercholesterolemia					
No	13	92.86	9	81.82	0.5648
Yes	1	7.14	2	18.18	
Hyperlipidemia					
No	8	57.14	10	90.91	0.0900
Yes	6	42.86	1	9.09	
Coronary artery disease					
No	12	85.71	11	100	0.4867
Yes	2	14.29	0	0	
Chronic renal failure					
No	13	92.86	11	100	1.0000
Yes	1	7.14	0	0	

Fisher exact test for categorical variables; Mann-Whitney U-test for continuous variables.

**Table 2 ijerph-19-03451-t002:** Comparison of mean values of ambient air pollutants and weather conditions between forest and urban parks.

Air Quality	N	Forest Park	N	Urban Park	*p*-Value
PM_1_ (μg/m^3^)	1129	19.59 ± 12.5	666	37.71 ± 26.6	<0.0001
PM_2.5_ (μg/m^3^)	1129	19.98 ± 12.6	666	38.35 ± 27.1	<0.0001
PM_10_ (μg/m^3^)	1129	21.17 ± 13.3	666	43.63 ± 43.3	<0.0001
Total PM (μg/m^3^)	1129	22.27 ± 14.6	628	86.96 ± 77.7	<0.0001
TVOC (ppb)	1026	157.08 ± 124.6	669	288.45 ± 99.5	<0.0001
CO (ppm)	1127	0.05 ± 0.01	668	2.11 ± 0.3	<0.0001
CO_2_ (ppm)	1127	419.95 ± 30.8	668	419.71 ± 13.0	0.8472
Temperature (°C)	1135	24.16 ± 1.5	336	34.38 ± 1.8	<0.0001
Humidity (%)	1135	87.05 ± 6.9	336	55.32 ± 4.7	<0.0001
Atmosphere (hPa)	144	886.44 ± 0.82	48	1000.76 ± 0.91	<0.0001
Wind speed (m/s)	144	1.18 ± 0.39	48	2.25 ± 1.15	<0.0001

**Table 3 ijerph-19-03451-t003:** Ambient air quality monitoring among five different forest trails.

Air Quality	*Phyllostachys edulis* Forest	*Japanese cedar* Forest	*Taiwan red* *cedar* Forest	Greensward	*Taiwania cryptomerioides* Forest	*p*-Value
	N = 208	N = 192	N = 191	N = 207	N = 188	
PM_1_ (μg/m^3^)	14.22 ± 2.9	19.04 ± 2.2	15.54 ± 3.6	25.14 ± 4.6	8.53 ± 2.5	<0.0001
PM_2.5_ (μg/m^3^)	14.70 ± 2.9	19.04 ± 2.2	15.83 ± 3.7	25.47 ± 4.6	8.78 ± 2.6	<0.0001
PM_10_ (μg/m^3^)	15.88 ± 3.1	19.89 ± 2.3	16.71 ± 3.9	26.60 ± 4.6	9.65 ± 2.8	<0.0001
Total PM (μg/m^3^)	16.70 ± 3.6	20.50 ± 2.5	17.39 ± 4.1	27.60 ± 5.0	10.41 ± 3.1	<0.0001
TVOC (ppb)	66.64 ± 39.1	66.14 ± 20.3	107.19 ± 17.0	58.08 ± 58.7	357.27 ± 32.1	<0.0001
CO (ppm)	0.11 ± 0.1	0.05 ± 0.01	ND	0.014± 0.01	0.03 ± 0.02	0.1594
CO_2_ (ppm)	412.56 ± 33.6	412.62 ± 26.2	418.30 ± 17.1	427.78 ± 46.3	423.86 ± 24.2	<0.0001
Temperature (°C)	23.88 ± 0.9	24.82 ± 1.9	22.77 ± 0.7	25.32 ± 1.7	24.16 ± 0.9	<0.0001
Humidity (%)	88.00 ± 4.5	84.32 ± 7.6	90.12 ± 5.0	81.84 ± 7.2	85.87 ± 5.3	<0.0001

ND: not detectable; N is the number of air quality monitoring per hour.

**Table 4 ijerph-19-03451-t004:** Ambient air quality monitoring among five urban parks.

Air Quality	228 Memorial Park	Chiang Kai Shek Memorial Hall Park	Pao An Temple Park	Sun Yat Sen Memorial Hall Park	Long Shan Temple Park	*p*-Value
	N = 145	N = 135	N = 124	N = 119	N = 143	
PM_1_ (μg/m^3^)	62.14 ± 39.2	36.50 ± 12.3	31.40 ± 6.9	30.63 ± 8.4	25.43 ± 26.1	<0.0001
PM_2.5_ (μg/m^3^)	63.16 ± 39.8	37.00 ± 12.4	31.94 ± 7.0	31.26 ± 8.6	25.94 ± 27.1	<0.0001
PM_10_ (μg/m^3^)	73.45 ± 61.6	39.01 ± 13.4	34.52 ± 7.4	35.14 ± 15.5	32.70 ± 57.9	<0.0001
Total PM (μg/m^3^)	147.00 ± 119.8	67.45 ± 56.2	64.61 ± 26.1	76.33 ± 81.2	88.05 ± 64.6	<0.0001
TVOC (ppb)	240.96 ± 53.2	352.61 ± 45.7	222.06 ± 48.4	186.86 ± 37.7	419.78 ± 47.1	<0.0001
CO (ppm)	2.31 ± 0.3	2.01 ± 0.1	2.28 ± 0.3	1.91 ± 0.1	2.00 ± 0.9	<0.0001
CO_2_ (ppm)	428.37 ± 8.6	414.00 ± 3.4	426.69 ± 14.3	401.65 ± 7.1	425.65 ± 6.0	<0.0001
Temperature (°C)	34.11 ± 2.1	36.31 ± 0.9	34.19 ± 1.7	34.28 ± 0.7	33.08 ± 1.1	<0.0001
Humidity (%)	58.42 ± 5.6	53.02 ± 1.3	50.64 ± 3.5	59.40 ± 2.7	55.06 ± 2.4	<0.0001

N is the number of air quality monitoring per hour.

**Table 5 ijerph-19-03451-t005:** Comparison of cardiovascular function before and after walking exercise in forest and urban parks.

Variables	Forest Park (76 Pairs)				Urban Park (86 Pairs)				*p*1-Value	*p*2-Value	*p*3-Value	*p*4-Value
	Before		After		Before		After					
	Mean	Std	Mean	Std	Mean	Std	Mean	Std				
**Blood Pressure Components (mmHg)**												
Systolic blood pressure	111.39	12.81	107.62	12.19	113.82	12.76	114.11	14.48	<0.0001	0.6927	0.2300	0.0026
Diastolic blood pressure	69.14	8.43	69.76	9.84	72.72	8.88	73.06	9.49	0.3951	0.4591	0.0098	0.0317
Central end systolic blood pressure	118.68	15.02	114.07	13.98	121.90	14.00	122.01	15.77	<0.0001	0.9135	0.1604	0.0009
Central end diastolic blood pressure	66.08	8.37	66.75	9.79	68.58	8.69	68.98	9.58	0.3545	0.4493	0.0653	0.1450
Mean artery pressure	83.58	9.65	82.22	10.49	85.17	9.17	85.74	10.43	0.0395	0.3556	0.2844	0.0343
Pulse pressure	52.61	10.64	47.32	8.88	53.33	8.74	53.02	9.05	<0.0001	0.7365	0.6371	<0.0001
**Cardiac Function**												
Heart rate (beat/min)	73.25	9.34	71.95	9.32	78.62	8.38	78.48	7.95	0.0131	0.7979	0.0002	<0.0001
Left ventricular ejection time (sec)	0.29	0.05	0.30	0.05	0.25	0.04	0.26	0.03	0.0025	0.0182	<0.0001	<0.0001
Left ventricular dP/dt max, (mmHg/s)	1123.9	211.4	1026.7	176.9	1212.3	185.2	1204.5	191.7	<0.0001	0.6536	0.0051	<0.0001
Left ventricular contractility (1/s)	15.56	1.70	14.81	1.59	16.21	2.17	16.29	1.82	<0.0001	0.7056	0.0344	<0.0001
Cardiac output (L/min)	5.04	1.00	4.80	0.97	5.22	0.92	5.24	0.83	<0.0001	0.7798	0.2343	0.0020
Cardiac index (L/min/m^2^)	2.87	0.49	2.72	0.45	3.04	0.57	3.05	0.48	<0.0001	0.9003	0.0374	<0.0001
Stroke volume (mL)	67.74	7.71	65.63	8.06	65.51	7.89	65.98	6.12	0.0007	0.5190	0.0720	0.7553
Stroke volume index (mL/m^2^)	38.59	3.74	37.38	3.88	38.09	4.69	38.35	3.54	0.0006	0.5308	0.4517	0.0991
**Vascular Function**												
Systemic vascular compliance (mL/mmHg)	1.34	0.28	1.43	0.27	1.26	0.18	1.28	0.18	0.0004	0.2782	0.0361	<0.0001
Systemic vascular resistance (dynes-sec/cm^5^)	1365.9	265.7	1414.5	286.2	1306.4	264.3	1340.6	260.3	0.0195	0.0534	0.1554	0.0869
Brachial artery compliance (mL/mmHg)	0.08	0.03	0.08	0.03	0.07	0.02	0.07	0.02	0.0010	0.7172	0.2255	0.0019
Brachial artery distensibility (%/mmHg)	6.60	1.69	7.41	1.92	6.27	1.17	6.34	1.17	0.0002	0.6267	0.1541	<0.0001
Brachial artery resistance (dynes-sec/cm^5^)	199.93	120.60	185.86	116.26	175.65	61.40	176.40	63.31	0.0132	0.8224	0.1163	0.5293

*p*1-value: paired *t*-test among forest parks; *p*2-value: paired *t*-test among urban parks; *p*3-value: two-sample *t*-test between forest parks and urban parks before walking; *p*4-value: two-sample *t*-test between forest parks and urban parks after walking.

**Table 6 ijerph-19-03451-t006:** Difference comparison of cardiovascular function changes after walking exercise in forest and urban parks.

Variables	Forest Park (76 Pairs)		Urban Park (86 Pairs)		*p*-Value
	Mean	Std	Mean	Std	
Systolic blood pressure (mmHg)	−3.78	7.45	0.29	6.80	0.0004
Diastolic blood pressure (mmHg)	0.62	6.30	0.34	4.28	0.7429
Central end systolic blood pressure (mmHg)	−4.62	8.69	0.10	8.91	0.0008
Central end diastolic blood pressure (mmHg)	0.67	6.28	0.41	4.97	0.7657
Mean artery pressure (mmHg)	−1.36	5.64	0.57	5.69	0.0324
Pulse pressure (mmHg)	−5.29	10.12	−0.30	8.31	0.0007
Heart rate (beat/min)	−1.30	4.47	−0.14	5.04	0.1242
Left ventricular ejection time (sec)	0.01	0.03	0.01	0.03	0.5460
Left ventricular dP/dt max, (mmHg/s)	−97.18	171.67	−7.85	161.72	0.0008
Left ventricular contractility (1/s)	−0.75	1.56	0.08	1.85	0.0024
Cardiac output (L/min)	−0.25	0.48	0.02	0.65	0.0038
Cardiac index (L/min/m^2^)	−0.15	0.27	0.01	0.38	0.0036
Stroke volume (mL)	−2.11	5.17	0.47	6.73	0.0074
Stroke volume index (mL/m^2^)	−1.22	2.94	0.26	3.86	0.0074
Systemic vascular compliance (mL/mmHg)	0.09	0.21	0.02	0.18	0.0271
Systemic vascular resistance (dynes-sec/cm^5^)	48.61	177.54	34.17	161.78	0.5889
Brachial artery compliance (mL/mmHg)	0.01	0.02	0.00	0.01	0.0084
Brachial artery distensibility (%/mmHg)	0.80	1.81	0.07	1.26	0.0035
Brachial artery resistance (dynes-sec/cm^5^)	−14.08	48.33	0.75	30.90	0.0235

Note: The difference = post-pre.

**Table 7 ijerph-19-03451-t007:** The coefficients of forest parks (vs. urban parks) using linear mixed models.

Outcome Variables	*β*	*p*-Value
Systolic blood pressure (mmHg)	−5.22	0.0004
Diastolic blood pressure (mmHg)	−0.12	0.9119
Central end systolic blood pressure (mmHg)	−6.91	0.0001
Central end diastolic blood pressure (mmHg)	0.59	0.5811
Mean artery pressure (mmHg)	−2.06	0.0662
Pulse pressure (mmHg)	−7.69	<0.0001
Heart rate (beat/min)	−2.46	0.0069
Left ventricular ejection time (sec)	0.03	<0.0001
Left ventricular dP/dt max, (mmHg/s)	−146.91	<0.0001
Left ventricular contractility (1/s)	−1.12	<0.0001
Cardiac output (L/min)	−0.52	<0.0001
Cardiac index (L/min/m^2^)	−0.28	<0.0001
Stroke volume (mL)	−3.02	0.0010
Stroke volume index (mL/m^2^)	−1.78	0.0006
Systemic vascular compliance (mL/mmHg)	0.16	<0.0001
Systemic vascular resistance (dynes-sec/cm^5^)	87.55	0.0029
Brachial artery compliance (mL/mmHg)	0.02	<0.0001
Brachial artery distensibility (%/mmHg)	1.67	<0.0001
Brachial artery resistance (dynes-sec/cm^5^)	−21.31	0.0033

Control variables: pre-test value, gender (male vs. female), age (54–73 y vs. 22–35 y), BMI (≥24 vs. <24), and hypertension (yes vs. no).

## Data Availability

The data presented in this study are available on request from the corresponding author.

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
