# Peer review of "Forest Bathing Is Better than Walking in Urban Park: Comparison of Cardiac and Vascular Function between Urban and Forest Parks"

_ijerph, 2022, doi:10.3390/ijerph19063451_

Round 1
Reviewer 1 Report
The paper has improved a lot, it is much more coherent than the previous version was.
I especially appreciated adding some factors that may affect CVFs in urban parks in the Discussion section, and extending the Limitations with the small sample size and varying participant characteristics.
As for the number of participants, I think I understand it now: 14 persons in the forest park + 11 persons in the urban park, but 4 in common, that is altogether 21 biologically but 25 mathematically/statistically (25 'research events').
Overall, I support publication of the manuscript.
Author Response
Response: We would like to thank you for the valuable comments and support publication of the manuscript.
Reviewer 2 Report
Dear Authors,
First, I would like to thank you for the revised manuscript and for taking into account my comments. Generally, the presented work, after corrections and additions, can be considered for publication. I also highly appreciate the originality of this research.
However, I still have minor comments/remarks on this version of the article.
- In the previous review, I asked you to refer to the various weather conditions in the park in the forest city park during the research. My comment in the previous review:
‘ The temperature and humidity conditions in Xitou Forest and the city parks were significantly different. In the case of air temperature, the difference between the forest and city parks was as much as 10 oC. In the case of relative humidity, the difference was approximately 30%. Could such different weather and humidity conditions have an impact on the results obtained in the parks? - I think that this aspect is worth mentioning, especially in the part of the discussion in which you write about the limitations of the study”.
In the revised version, the authors, unfortunately, wrote only one sentence on this topic (in the discussion).
„Fourth, the weather conditions including air temperature, atmospheric pressure and wind speed in urban parks have not been considered.”
I think at least the word "different" should be added here, e.g. “Fourth, the different weather conditions, including air temperature, atmospheric pressure and wind speed, that occurred during the research in urban and forest parks have not been considered”
- The title of table 2 should be completed with words (marked in bold)
Comparison of the mean values of ambient air quality monitoring and weather conditions between forest and city parks, which occurred during the research.
In my opinion, this should be corrected before considering the article for publication.
Best regards,
Author Response
I would like to thank for your inspiring comments and suggestions.
Please see the attached file about author's reply.

Reviewer 3 Report
This is an interesting work on the benefits of forest bathing on human health and in particular on cardiovascolar and other related functions in forested aread compared with urban parks.
It could be accepted after the following revisions:
-provide more literature references for forest bathing practice at its first mention in the Introduction
-the conclusion at pag 4 that 'that forest parks have better air quality compared to that in Taipei urban parks' should be enriched by hypothesizing that also levels of biogenic volatile organic compounds with healthy potential inhalable during sport are higher in forests than urban parks. Also, noise/silence conditions are different in the two different environments. Report this concept also in Discussion.
-consider for a future work to monitor not only TVOC but separately also biogenic VOCs (BVOCs) and anthropogenic VOCs (AVOCs)
-The revised Introduction should contain a short mention of the combined effects of forest bathing together with sport on human health. Breathing volatile compounds from plants (biogenic VOCs), especially from old-growth forests, can have effects on human health and thus 'forest bathing' is considered a beneficial practice against COVID-19-related disorders to be accompanied by sport, especially walking in a forest. (DOI: 10.1007/s10311-021-01321-9; 10.1007/s10311-021-01372-y) Even indoor plants in houses and offices can help people to bolstering their immune system (DOI:10.3390/ijerph19010273 ) Comment on the above points and cite at least the works indicated by DOI above.
-sec. 2.1 what was the average temperature outside during the forest therapy studies? How many sunlight hours/day?
-sec. 2.1: provide city, province, country for parks used during the studies at any mention of them
-sec. 2.1: provide city and country for NTU Hospital at any mention of it
-walkers in urban parks can be disturbed also by city noises add after ref 35, pag 13
-pag 14: 'Second, exposure to forest environments showed some beneficial effects on NK cells,': specify: ' environments and plant-released biogenic VOCs inhaled by walkers, showed....'
-page 14: 'Forth, Better green landscape ': better should be not in capital letters
-consider to give Limitations and Strengths of the study in a separate section
-This is a small-sized study. Provide literature references of works published on the basis of similar small numbers of participants
-in legend of Tab 3 scientific names of plants mentioned could be given
Author Response
I would like to thank for your valuable comments and suggestions.
Please see the attached file about author's reply.

This manuscript is a resubmission of an earlier submission. The following is a list of the peer review reports and author responses from that submission.
Round 1
Reviewer 1 Report
The paper: “Forest bathing is Better Than Walking in Urban Park: Comparison of Cardiac and Vascular Finction Between Urban and Forest Parks” present interesting idea, however the experiment itself has been incorectly planned. Investigated groups differ in average age about 10years (40 to 50average) which in this case can be consider as serious flaw. And what is even more some of them had cardiovascular diseases and others not which makes any inference highly complicated. The groups are to heterogenous and too small in the same time to consider them as reliable. Also the the unequal share of women and men raises considerable doubts. Moreover the time of the undertaken walks was different between groups (the forest group had 1.5h walk, and the second group 1.5h walk in each of 5 parks – this is highly misleading). Moreover, the investigated parks had not been described or analyzed with taking into consideration their surface area, green coverage or plants groups in them. It is a crucial information in this case if those parks includes high number of trees or no trees and shrubs at all. It is also a big drawback taht there was no controll group walking trough the urban area to investigate if the parks or forests improve the health condition in compare to urban area itself. Also the air pollution parameters should be compared to the urban area, because it is obvious that the pollution level would be smaller in the forest than in urban park. The conclusions can lead to a harmful idea, that walking trough urban parks has no advantagous effect on human health. Maybe it should be underlined if those parks there was a shortage of plants (especially big ones) and for this reason more big trees should be planted in cities to improve health of the citizens. However this is more a hint for future studies, while such conclusions cannot be drawn from improperly planned experiment.
Reviewer 2 Report
Please see the attached document.

Reviewer 3 Report
Dear authors,
First, I would like to thank you for presenting your results. Overall, I found that the work is very interesting. In my opinion, the research was done well. The presented data in the manuscript are new, the objectives of the paper are clear. The study design setup and analysis are good. The article is understandably written and well-organized, contain all the components I would expect, and the sections are well-developed. The methodology is clearly explained, the results are well described, and the discussion is carried out well. In my opinion, good paper.
However, I have some minor comments, mainly to sections Material and Methods and Discussion.
- Lines 89-91 – It seems to me that the second sentence (lines 90-91) is an unnecessary repetition of the previous one.
- I am not very good at medical experience, but the research in the forest lasted 6 days, while the study in city parks was only two days. I think you should refer to this in the discussion.
- Lines 122-123- “ The mean temperature and relative humidity from 2011 to 2015 were 17 °C and 88%.
The given values should be completed and extended. I suppose that the given values are average annual values of climatic parameters, but this should be written in the article. I would also suggest using the term “air temperature” ( e.g. “ The average annual air temperature and relative humidity from 2011 to 2015 were 17 °C and 88% ).
Moreover, the values given are for Xitou Forest only, no such information is available for city parks. So I believe that these values ​​should be supplemented.
I also have some comments regarding the characteristics and analysis of the weather conditions during your research.
- The temperature and humidity conditions in Xitou Forest and the city parks were significantly different. In the case of air temperature, the difference between the forest and city parks was as much as 10 oC. In the case of relative humidity, the difference was approximately 30%. Could such different weather and humidity conditions have an impact on the results obtained in the parks? - I think that this aspect is worth mentioning, especially in the part of the discussion in which you write about the limitations of the study.
- I also think that it was worth in the article (in Table 2) to include information about the barometric (atmospheric pressure) and wind conditions on the days of the research. To my knowledge, these conditions can have an impact on cardiac and vascular function. It would also be important to check that there were no large changes in atmospheric pressure from day to day on the days of your experiment. At the end of August, such a situation may occur in northern Taiwan (city parks). Maybe it would also be worth checking the air conditions (speed, direction). If there were large changes in atmospheric pressure from day to day as well as different wind conditions, it could have had an impact on the results in city parks.
In the discussion, in the section where you discuss the limitations of your study, you should indicate that the results obtained may have been influenced by the different lengths of the study period and different weather conditions at both test sites.
I would also suggest that in your next study on the influence of the forest on Cardiac and Vascular Function, you should include indicators in the field of human bioclimatology.
Best regards,
Round 2
Reviewer 1 Report
After reading the improved version of the manuscript I maintain my position on rejection the paper. This paper has low scientific value, it was improved in some point accroding to suggestions, however the design of the experiment is difficult to accept. The two compared groups are in fact different people, and what is even more those groups are highly heterogenous concerning age, sex and diseases. Such groups cannot be compared in such combination. Treatements were also not unifromed, and many other factors can interfere with the tested ones. The investigated parks differ greatly in covered surfaces and cannot be treated as one homogenous group. Parks with vegetative areas 117,340 m2 and 5,913 m2 cannot be treated as same "urban park area" while many parameters such as temperature, shade, noise and others would differ greatly (and all of those can alter the measured parameters). The conclusion that besides "sample size was relatively small" ... "large amount of repeated measurements... can increase the statistical power to detect the true difference of health effects in different environments" is in opposite to the science of statistics and with scientific attitude in general. The last paragraph of the 'Discussion' section gather all the 'limitations' which are for me sum of deficiencies disqualifying the paper for publication in such good journal as Yours.